# Systematic generalisation with group invariant predictions

**Faruk Ahmed**[1]*, **Yoshua Bengio**[1,2]**, Harm van Seijen**[3]**, Aaron Courville**[1,2]
[1] Université de Montréal, Mila, [2]CIFAR Fellow, [3]Microsoft Research

## Abstract

We consider situations where the presence of dominant simpler correlations with the target variable in a training set can cause an SGD-trained neural network to be less reliant on more persistently correlating complex features. When the non-persistent, simpler correlations correspond to non-semantic background factors, a neural network trained on this data can exhibit dramatic failure upon encountering systematic distributional shift, where the correlating background features are recombined with different objects. We perform an empirical study on three synthetic datasets, showing that group invariance methods across inferred partitionings of the training set can lead to significant improvements at such test-time situations. We also suggest a simple invariance penalty, showing with experiments on our setups that it can perform better than alternatives. We find that even without assuming access to any systematically shifted validation sets, one can still find improvements over an ERM-trained reference model.

## 1 Introduction

If a training set is biased such that an easier-to-learn feature correlates with the target variable throughout the training set, a modern neural network trained with SGD will use that factor to perform predictions, ignoring co-occurring harder-to-learn complex predictive features (Shah et al., 2020). Without any other criteria, this is arguably desirable behaviour, reflecting Occam's razor. We consider the situation where although such a simpler correlation is a dominant bias in the training set, a minority group exists within the dataset where the bias does not manifest. In such cases, relying on more complex predictive features which more pervasively explain the data can be preferable to simpler ones that only explain most of it. For example, if all chairs are red, redness ought to be a predictive rule for chairhood (without any other criteria for predictions). However, if some chairs are not red, and all chairs have backs and legs, then one can infer that redness is less relevant.

In this paper, we will study object recognition tasks, where the objects correlate strongly with simpler non-semantic background information for a majority of the images, but not for a minority group. There is evidence in the literature that modern CNNs tend to fixate on simpler features such as texture (Geirhos et al., 2019; Brendel & Bethge, 2019), canonical pose (Alcorn et al., 2019), or contextual background cues (Beery et al., 2018). We are assuming that semantic features in a classification context (ones that humans would agree contribute to their labelling of objects) are more likely to persistently correlate with the target variable, while simpler non-semantic background biases are more likely to exhibit non-persistent correlations in real-life data collection processes. Based on this assumption, we will use combinations of objects and backgrounds to compare test-time performances corresponding to particular distributional shifts.

Consider coloured MNIST digits such that there is a dominant, but not universal, correlation between colour and digit identity for a majority of the images. In the situation we are considering, if the biasing colours in the majority group are not recombined with different digits in the minority group, then there is no signal for the model to disregard these biasing factors, which are retained as important predictive rules. This can lead to poor performance at *systematic generalisation* (Lake & Baroni, 2018), where an object occurs with another object's biasing factor, and at *semantic anomaly detection* (Ahmed & Courville, 2020), where a novel object appears with one of the biasing factors. In our example

---

*Correspondence to faruk.ahmed@umontreal.ca.

Table 1: For a coloured MNIST dataset with every digit correlated with a colour 80% of the time, we see poor performance at systematically varying tasks. Performance improves if the minority group combines colours from other biased digits - this provides corrective gradients that promote invariance to colour. Non-systematic shifts are when unseen colours are used, and anomaly detection is measured by decreased predictive confidence for an unseen digit (see Section 2 for more details).

| Minority colours | In-distribution | Non-systematic shift | Systematic shift | Anomaly detection |
|---|---|---|---|---|
| Different | $99.60 \pm 0.02$ | $53.26 \pm 1.89$ | $38.72 \pm 2.27$ | $7.70 \pm 0.23$ |
| Recombinations | $98.67 \pm 0.39$ | $85.05 \pm 1.89$ | $97.56 \pm 0.05$ | $46.59 \pm 6.93$ |

with coloured MNIST, if we colour the minority group digits with the colours used to bias (different) digits in the majority group, we find a marked improvement at systematically shifted tests over the case when the colours in the minority group are different colours altogether (see Table 1).

We investigate the role of encouraging robust predictive behaviour across such groups in terms of improved performance at tasks with such distributional shifts. Our experiments suggest that training with cross-group invariance penalties can result in models that have learned to be more reliant on persistent complex correlations without being overwhelmed by simpler, yet less stable features, as indicated by improved performance at systematic generalisation and semantic anomaly detection on our synthetic setups.

We find that a recently proposed method (Creager et al., 2020) can be effective at inferring the majority and minority groups along a learned feature-bias, and we use this inferred partition to provide us with groups in the training set in our comparative study. We also suggest a new method for encouraging predictions that rely on persistent correlations across such groups, with the intuition that similar predictive behaviour across the groups should be promoted throughout training. With experiments on three synthetic datasets, we compare the performance of recently proposed invariance penalties and methods, and find that our variant can often perform better at tasks involving such test-time distributional shifts.

## 2 SYSTEMATIC AND NON-SYSTEMATIC GENERALISATION

If we assume that data $x$ is generated via a composition $\mathcal{C}$ of semantic factors $h_s$ and non-semantic factors $h_n$, we can use this decomposition, $x = \mathcal{C}(h_s, h_n)$, to generate test datasets to capture different scenarios. While $h_n$ is actually independent of $y$, we shall have the independence property $p_{\mathcal{D}}(h_n|y) = p_{\mathcal{D}}(h_n)$ to not hold when there is bias in the dataset $\mathcal{D}$ due to $h_n$–$y$ correlations.

We can evaluate, for a particular target $y$ and our system's prediction of the target $\hat{y}(x)$, the average accuracy $\mathbb{E}\big[\mathbf{1}\{\hat{y}(\mathcal{C}(h_s, h_n)) = y\}\big]$, as a measure of generalisation for the following different cases.

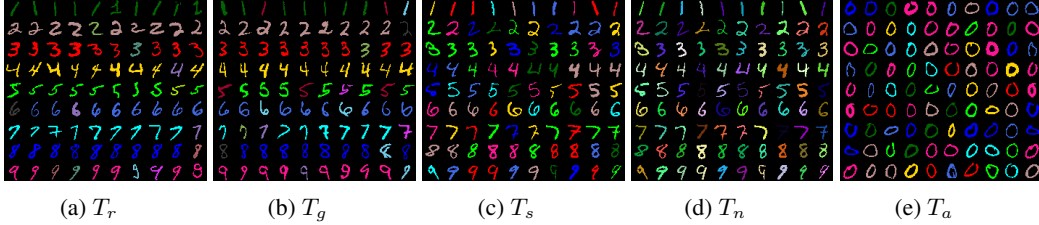

| (a) $T_r$ | (b) $T_g$ | (c) $T_s$ | (d) $T_n$ | (e) $T_a$ |
|---|---|---|---|---|

Figure 1: COLOURED MNIST training and test sets for evaluating generalisation under non-semantic marginal shift and systematic shift, and anomaly detection. (a) Training set; (b) *In-distribution generalisation set $T_g$*, where the test set is coloured following the same scheme as for $T_r$; (c) *Systematic-shift generalisation set $T_s$*, where we colour the test set with the biasing colours, but such that no digit is coloured with its own biasing colour; (d) *Non-systematic-shift generalisation set $T_n$*, where the test is coloured with random colours that are different from any of the colours seen in the training set; and (e) *Semantic anomaly detection set $T_a$*, where we colour the held-out digits of the test set randomly with the biasing colours.

**In-distribution generalisation** $h_s \sim p(h_s|y)$ and $h_n \sim p(h_n|y)$: The validation and test sets are assumed to possess the same biases as the training set, in that the class-conditional distribution of non-semantic features in the test set match that of the training set, $p(h_n|y)$.

**Generalisation under non-systematic-shift** $h_s \sim p(h_s|y)$ and $h_n \not\sim p(h_n)$[1]: This estimates a form of generalisation under distributional shift, where the non-semantic factors are sampled from outside the marginal distribution of $h_n$ as present in the training set.

**Generalisation under systematic-shift** $h_s \sim p(h_s|y)$ and $h_n \sim p(h_n|y')$ where $y' \sim p(y)$ $s.t.$ $y' \neq y$: This estimates another form of generalisation under distributional shift but one where non-semantic factors are sampled with intent to confuse: non-semantic factors for $x$ are sampled from the marginal distribution of a randomly picked different target, $y' \neq y$. Although systematicity, as discussed in Fodor & Pylyshyn (1988), and systematic generalisation, as discussed in the NLP literature (Lake & Baroni, 2018; Bahdanau et al., 2019) consider recombinations of intra-semantic factors as well, here, in the context of background-agnostic object recognition tasks, we only consider $h_s - h_n$ recombinations.

**Semantic anomaly detection** $h_s \not\sim p(h_s)$ and $h_n \sim p(h_n)$: Such a datapoint should not be confidently categorised as a known $y$, even if non-semantic features are shared (Ahmed & Courville, 2020). We can use these $x$ to evaluate anomaly detection, as indicated by decreased predictive confidence, and measured by the area under the precision-recall curve (Hendrycks & Gimpel, 2017).

COLOURED MNIST: Consider an illustrative dataset with coloured MNIST digits. For the training set, $T_r$, MNIST digits are coloured with a set of digit-correlated "biasing" colours 80% of the time, and with ten random colours that are different from the biasing colours the remaining 20% of the time. One digit is held out, for testing semantic anomaly detection. See Figure 1 for examples of the four test sets corresponding to this setting, and also Appendix A for more details on the construction.

Improving performance for such scenarios involving distributional shift might come at a cost for in-distribution performance, since more robust features might be harder to learn than simpler dominant correlations that hold in-distribution. In real-world deployments where one is likely to encounter unexpected situations, such as in a self-driving car, it can often be preferable to find appropriate trade-offs such that classifiers can indicate reduced confidence upon encountering anomalous objects, or continue to operate in changing environments, while continuing to achieve a desirable degree of in-distribution predictive performance.

## 3 PREDICTIVE GROUP INVARIANCE ACROSS INFERRED SPLITS

In general, we do not expect to have direct knowledge of majority and minority groups corresponding to the biasing non-semantic features in a dataset. We will later show how one might infer such groups from the data, but we first describe an invariance penalty assuming we have access to the groups.

Learning features that are group invariant would require us to match the (class-conditioned) distribution of features from the majority and minority groups (Ganin et al., 2016; Li et al., 2018a). In terms of predictive performance, we can alternatively ask for the class-conditioned distributions of features to match in the sense that they lead to the same softmax distributions on average as training progresses, without modifying the last linear layer. This implementation has the advantage of doing away with an adversarial network, and the issues that tend to accompany the training of such models. We shall refer to this objective as *predictive group invariance* (PGI). Intuitively, encouraging matched predictive distributions across the groups with a fixed last layer pushes for over-emphasis on minority-group features in the representation, thus acting as an implicit re-weighting of features in both groups (leading to demoting the relevance of colour in the MNIST case, for example). When a persistent feature does exist in both groups, using that feature can lead to equal training rates in regularised networks, satisfying the penalty.

Consider a classifier that extracts a feature vector $f_\theta(x)$, where $\theta$ are the parameters of a convolutional neural network for example, with a linear layer $w$ on top. The predictive distribution is then

$$p_w(y|x) = \sigma(w^\top f_\theta(x)), \tag{1}$$

where $\sigma$ is a softmax, and predictions are made by performing an $\mathrm{argmax}$.

---

[1]In this paper, we imply sampling from outside the support of $p$ when we say $h \not\sim p(h)$.

Given a partition scheme for splitting the images $x$ in our dataset $\mathcal{D}$ such that every $i$-th image $x^{(i)}$ is associated with a partition-label $\alpha^{(i)}$, we define distributions $\mathbb{P}^c, \mathbb{Q}^c$ for the subsets in class $c$:

$$x^{(i)} \sim \mathbb{P}^c \text{ if } \alpha^{(i)} = 0, y^{(i)} = c, \tag{2}$$

$$x^{(j)} \sim \mathbb{Q}^c \text{ if } \alpha^{(j)} = 1, y^{(j)} = c. \tag{3}$$

We want to minimize empirical risk under the constraint that our feature extractor causes similar predictive distributions on average for pictures of the same object in both partitions. Formally, we want to optimise

$$\min_{\theta, w} \ell(\theta, w | \mathcal{D}), \tag{4}$$

$$\text{s.t. } \theta \in \underset{\Theta}{\operatorname{argmin}} \, d\Big( \underset{x \sim \mathbb{P}^c}{\mathbb{E}}[p_w(y|x)], \underset{x \sim \mathbb{Q}^c}{\mathbb{E}}[p_w(y|x)] \Big), \ \forall \, c, \tag{5}$$

where $\ell$ is the standard loss function for ERM training, for example, the categorical cross-entropy. A softened objective for stochastic optimisation can be approximated as

$$L(w, \theta | \mathcal{D}, \alpha) = \ell(\theta, w | \mathcal{D}) + \lambda \Bigg[ \sum_c d\Big( \underset{x \sim \mathbb{P}^c}{\mathbb{E}}[p_{\tilde{w}}(y|x)], \underset{x \sim \mathbb{Q}^c}{\mathbb{E}}[p_{\tilde{w}}(y|x)] \Big) \Bigg]_{\tilde{w}=w \text{ (fixed)}}. \tag{6}$$

Since we are comparing distributions, we make the simplest natural choice of $d$ to be the KL-divergence,

$$d\Big( \underset{x \sim \mathbb{P}^c}{\mathbb{E}}[p_{\tilde{w}}(y|x)], \underset{x \sim \mathbb{Q}^c}{\mathbb{E}}[p_{\tilde{w}}(y|x)] \Big) = \sum \underset{x \sim \mathbb{Q}^c}{\mathbb{E}}[p_{\tilde{w}}(y|x)] \log \frac{\mathbb{E}_{x \sim \mathbb{Q}^c}[p_{\tilde{w}}(y|x)]}{\mathbb{E}_{x \sim \mathbb{P}^c}[p_{\tilde{w}}(y|x)]}. \tag{7}$$

We use this particular ordering of $\mathbb{Q} \| \mathbb{P}$ because with our grouping, $\mathbb{P}$ consists of examples that are "easy" due to a particular bias, and so the mean predictive distribution for $\mathbb{P}$ tends to be correct and low-entropy, while that for $\mathbb{Q}$ is more high-entropy and inaccurate. We take advantage of the zero-forcing property of this KL divergence, encouraging the mean predictive distribution for $\mathbb{Q}$ to closely match that of $\mathbb{P}$. It is likely that different choices for $d$ would be better suited for different settings.

**Partitioning the dataset** Recently, Creager et al. (2020) have considered the question of finding worst-case partitions for invariant learning given a collection of data. The key intuition is that an invariant learning objective, as formulated by IRM (Arjovsky et al., 2019), is maximally violated by splitting along a spurious correlation when predictions rely exclusively on it in a reference model (see Theorem 1 in Creager et al. (2020) for details). In our case, this would consist of partitioning into the majority and minority groups given our ERM-trained model early on in training as reference.

A soft-partition predicting network is used, $g(x, y)$, conditioned on the input and the target, to *maximise* the IRMv1 penalty (Arjovsky et al., 2019), which gives us soft partition-predictions, $\hat{\beta}$, for the examples,

$$\hat{\beta} = \max_{\beta} \sum_{e \in \{0,1\}} \frac{1}{\sum_{i'} \beta^{(i)}(e)} \sum_i \beta^{(i)}(e) \ell(\sigma(\Phi(x^{(i)})), y^{(i)})$$

$$+ \sum_{e \in \{0,1\}} \gamma \Big\| \nabla_{\mu | \mu = 1.0} \frac{1}{\sum_{i'} \beta^{(i)}(e)} \sum_i \beta^{(i)}(e) \ell(\sigma(\mu \circ \Phi(x^{(i)})), y^{(i)}) \Big\|^2, \tag{8}$$

where $\Phi(x_i) = w^\top f_\theta(x)$ are the logits from the reference model, $e \in \{0, 1\}$ indexes the partition, $\beta^{(i)}(e) \in [0, 1]$ signifies the predicted probability for the $i$-th example being in partition $e$, such that $\beta^{(i)}(e = 0) + \beta^{(i)}(e = 1) = 1$, and $\gamma$ is a hyper-parameter. We can then compute the partition $\alpha^{(i)} = \operatorname{argmax}_e \beta^{(i)}(e)$. In our implementation, we condition the partition predicting network $g$ on the features $f_\theta(x)$ instead of the input $x$, and use separate networks for each category, *i.e.* $\beta^{(i)} = g_{y^{(i)}}(f_\theta(x^{(i)}))$. We find this to perform better in preliminary experiments, improving training and enabling more light-weight $g$ networks. This also ensures that the same features as the ones used by our ERM-trained reference model are used to predict partitions, resulting in partitions corresponding to more consistent learned-feature biases. We provide more details in Appendix B.3.

Figure 2: (left) COCO-ON-COLOURS; left block is the majority group, right block is the "unbiased" minority group; (right) COCO-ON-PLACES.

## 4 RELATED WORK

The dominant perspective towards the issue of unreliable behaviour in novel domains has consisted of treating the problem as that of *domain generalisation* (Blanchard et al., 2011). One hopes to recover stable features by encouraging invariance across data sampled from different domains, so that performance at test-time *out-of-distribution* (OoD) scenarios is less likely to be unstable.

Approaches along such lines typically resemble a cross-domain distribution-matching penalty applied to the features being learned, augmenting the usual ERM term (Ganin et al., 2016; Sun & Saenko, 2016; Heinze-Deml & Meinshausen, 2017; Li et al., 2018; Li et al., 2018a;b), and evaluated on datasets that consist of data in different modalities (Li et al., 2017; Peng et al., 2019; Venkateswara et al., 2017), or collected through different means (Fang et al., 2013), or in different contexts (Beery et al., 2018).

Works with the perspective of *distributionally robust optimisation* (DRO) have generally considered using uncertainty sets around training data (Ben-Tal et al., 2013; Duchi & Namkoong, 2018) to minimise worst-case losses, which can often have a regularising effect by effectively up-weighting harder examples. More relevant to our discussion, group DRO methods have considered uncertainty sets in terms of different groups of data, for example with different cross-group distributions of labels (Hu et al., 2018), or groups collected differently (Oren et al., 2019), similarly to domain generalisation datasets.

More recently, methods promoting the learning of stable features across data from different *environments*, or sources, have been proposed by using gradient penalties (Arjovsky et al., 2019), risk-based extrapolation (Krueger et al., 2020), and masking gradients with opposing signs (Parascandolo et al., 2020).

The typical datasets in such existing works are not curated with testing performance under systematic distributional shift in mind, most often not characterising the specific shift in distribution. In recent times, a commonly adopted synthetic dataset is the coloured MNIST variant used in Arjovsky et al. (2019) – since this particular dataset uses flipped colours for the minority group, which is less of a problem with ERM-training, the true digit labels were flipped at a sufficiently high frequency to incapacitate ERM performance by forcing reliance on colour. We believe setups such as ours can be better synthetic testbeds for developing ideas, where it is not necessary to alter ground truth labels to expose a failure mode. In general, using better models of dataset bias implies a narrower disconnect with realistic settings, with higher chances of the conclusions carrying over.

## 5 EXPERIMENTS

We compare performance with our four test sets - in-distribution, non-systematically shifted, systematically shifted, semantic anomalies - for a range of recently proposed methods for a set of three synthesised datasets. Appendix B describes architectural details and training choices.

### 5.1 METHODS

We compare recent methods aimed at robust predictions across groups, and which do not require changes to network capacity or additional adversaries to impose invariance penalties. We also do not include methods based on advances in self-supervised feature learning, such as Carlucci et al.

(2019), since such methods are developed with prior knowledge of the desired invariances, and are thus limited in their generality.

**Baseline**: This is our reference model, trained via ordinary (regularised) empirical risk minimisation (ERM) without any invariance penalties added. The choices for architecture and regularisers were made to conform to the way modern networks are typically trained with in-distribution performance in mind (details in Appendix B).

**IRMv1**, **REx**, **GroupDRO**: IRMv1 (Arjovsky et al., 2019) and REx (Krueger et al., 2020) are two methods that augment the standard ERM term with invariance penalties across data from different sources. GroupDRO (Sagawa et al., 2020) is an algorithm for distributional robustness, which works by weighting groups of data as a function of their relative losses. See Appendix C for more details about these methods.

**cIRMv1**, **cREx**, **cGroupDRO**: We implement label-conditional variants of the above algorithms, which, to our knowledge, has not been explored. In the context of multi-class classification it is reasonable to expect that performances might have multi-modal distributions along different categories earlier in training, which suggests stratification by class might improve performance.

**Reweight**: We weight the losses in the biased group down. This is a heuristic form of re-balancing the dataset, while choosing a hyper-parameter for the weight using the validation set, with the weight serving to downweight the losses for the biased group. In preliminary experiments we found this re-weighting variant (King & Zeng, 2001) to significantly outperform oversampling the minority group, as suggested in Buda et al. (2018), or weighting the grouped losses using their population ratios, as performed for imbalanced classes in Cui et al. (2019).

**cMMD**: Following Li et al. (2018), we match the MMD (Gretton et al., 2012) of the distribution of features. In preliminary experiments, we find a conditional version (as done with adversarial models in (Li et al., 2018a)) to perform significantly better, so we only report cMMD results here.

## 5.2 DATASETS

Evaluating performance in an unambiguous manner for the specific kinds of generalisation that we aim to study necessitates controlled test-beds. In order to model these tasks, we use 3 synthetic datasets of progressively higher complexity, approaching photo-realism.

COLOURED MNIST: This is the simplest setting, where the background information exists as part of the object.

COCO-ON-COLOURS: We superimpose 10 segmented COCO (Lin et al., 2014) objects on coloured backgrounds. The training set has 800 images per category, with nine in-distribution categories and one held-out category for anomaly detection. Validation and test sets have 100 each images per category. See Figure 2 (left). This is the most extreme dataset in our experiments in terms of the contrast in complexity between the non-semantic correlating factor (background colour) vs. stable features (objects).

COCO-ON-PLACES: Here we superimpose the same COCO objects on scenes from the PLACES dataset (Zhou et al., 2017), with the place-scenes acting as the bias (figure 2, right). See Appendix A for more details about how these datasets are constructed. While the backgrounds in this dataset are more complex than colour, they still act as biasing factors, as indicated in the relatively poorer performance at systematic generalisation, and were selected due to visually obvious and distinct colour or texture.

## 5.3 RESULTS

In all cases, we have used the partition predictor to infer the two groups. The partition accuracies for the three datasets at the end of one epoch of training the base models are in the table below. We tested a more naïve approach by applying K-Means clustering to the losses, but found it to under-perform, possibly because it cannot account for a consistent feature bias learned by our reference model.

| COLOURED MNIST | COCO-ON-COLOURS | COCO-ON-PLACES |
|:---:|:---:|:---:|
| $97.26 \pm 0.71$ | $98.22 \pm 1.05$ | $80.43 \pm 1.41$ |

Table 2: Generalisation results on COLOURED MNIST.

| Methods | In-distribution | Non-systematic shift | Systematic shift | Anomaly detection |
|---|---|---|---|---|
| Base (ERM) | **99.60 ± 0.02** | 53.26 ± 1.89 | 38.72 ± 2.27 | 7.70 ± 0.23 |
| IRMv1 | 99.47 ± 0.05 | 63.24 ± 3.04 | 55.19 ± 1.07 | 11.54 ± 1.18 |
| REx | 98.95 ± 0.11 | 72.12 ± 1.90 | 71.18 ± 3.27 | 15.54 ± 2.05 |
| GroupDRO | 89.47 ± 4.52 | 70.53 ± 1.79 | 79.17 ± 1.64 | 35.15 ± 10.83 |
| Reweight | 98.51 ± 0.12 | 75.01 ± 1.28 | 84.85 ± 0.61 | 28.60 ± 1.11 |
| cIRMv1 | 99.36 ± 0.25 | 65.78 ± 3.53 | 61.09 ± 5.30 | 14.16 ± 2.12 |
| cREx | 98.56 ± 0.12 | 74.35 ± 2.09 | 80.01 ± 2.11 | 22.02 ± 2.52 |
| cGroupDRO | 95.65 ± 3.23 | 75.41 ± 3.45 | 81.14 ± 2.41 | 26.61 ± 6.61 |
| cMMD | 99.40 ± 0.03 | 97.17 ± 0.59 | 97.86 ± 0.16 | 78.32 ± 4.15 |
| PGI | 99.05 ± 0.08 | **98.58 ± 0.06** | **98.48 ± 0.05** | **89.42 ± 1.95** |

Table 3: Generalisation performance on COCO-ON-COLOURS.

| Methods | In-distribution | Non-systematic shift | Systematic shift | Anomaly detection |
|---|---|---|---|---|
| Base (ERM) | 90.57 ± 1.28 | 26.81 ± 4.93 | 1.10 ± 0.36 | 5.47 ± 0.08 |
| IRMv1 | 91.61 ± 0.38 | 32.30 ± 4.52 | 2.11 ± 0.30 | 5.81 ± 0.17 |
| REx | **91.69 ± 0.50** | 36.57 ± 4.03 | 2.69 ± 0.81 | 5.73 ± 0.14 |
| GroupDRO | 43.06 ± 2.26 | 41.32 ± 4.39 | 43.24 ± 2.89 | 20.05 ± 3.08 |
| Reweight | 42.42 ± 3.47 | 47.56 ± 2.27 | 49.12 ± 1.63 | 18.15 ± 3.81 |
| cIRMv1 | 91.53 ± 0.31 | 31.11 ± 4.51 | 1.74 ± 0.40 | 5.87 ± 0.16 |
| cREx | 74.75 ± 14.14 | 32.29 ± 7.71 | 29.75 ± 5.16 | 19.77 ± 14.98 |
| cGroupDRO | 41.10 ± 2.37 | 41.83 ± 2.96 | 42.10 ± 2.15 | **21.81 ± 5.40** |
| cMMD | 89.87 ± 1.13 | 55.02 ± 2.29 | 27.36 ± 1.57 | 8.82 ± 0.70 |
| PGI | 78.23 ± 2.01 | **55.57 ± 4.60** | **51.62 ± 3.09** | 18.84 ± 2.11 |

In Tables 2,3,4, we find that significant improvements can be achieved using group invariance methods. All hyper-parameters for the results in this set are picked on a validation set consisting of a subset of colours or backgrounds that are different from both the training and test sets, and an equally sized subset of systematically varying colours or backgrounds from the biased majority group. In all cases, the split is learned after one epoch of training, and the various penalties dropped in at this point with a linearly ramped-in penalty co-efficient. Details about hyper-parameter selection are in Appendix C.

While conditional variants perform better at systematic generalisation for COLOURED MNIST, perhaps owing to our hyper-parameter selection procedure of using a mixed-shift validation set, performance at systematic shift appears to be traded off with non-systematic shift in some cases for the more complex datasets. All aggregates are over 5 trials.

## 5.4 PRACTICAL CONSIDERATIONS FOR HYPER-PARAMETER SELECTION

While we find that with the use of group invariance penalties it is possible to encourage reliance upon complex persistent correlations in the presence of dominant simple biases, this can sometimes come at a cost to in-distribution performance when picking hyper-parameters using validation sets with specific distributional shift. One might reasonably expect that this can be mis-aligned with real-life situations: in practice, one typically does not have access to data corresponding exactly to unexpected scenarios, besides not expecting to encounter situations outside the training distribution nearly as often as situations for which a model has been trained and deployed. A practitioner might wish to aim for a clearer trade-off with such situations, with prior knowledge of how often they might arise compared to in-distribution situations, and with a surrogate validation set to model distributional shift. Here, we will simply show that picking hyper-parameters without assuming access to validation sets consisting of systematic distributional shift can still provide improvements over the baseline reference model. We consider three cases.

Table 4: Generalisation performance on COCO-ON-PLACES.

| Methods | In-distribution | Non-systematic shift | Systematic shift | Anomaly detection |
|---------|-----------------|---------------------|------------------|-------------------|
| Base (ERM) | $81.06 \pm 1.01$ | $45.25 \pm 0.96$ | $29.18 \pm 1.24$ | $9.21 \pm 0.21$ |
| IRMv1 | $80.93 \pm 0.71$ | $45.17 \pm 0.92$ | $28.78 \pm 0.73$ | $9.39 \pm 0.60$ |
| REx | $\mathbf{81.55 \pm 0.70}$ | $45.35 \pm 0.92$ | $29.56 \pm 0.77$ | $9.46 \pm 0.51$ |
| GroupDRO | $76.05 \pm 0.87$ | $43.72 \pm 0.43$ | $31.83 \pm 0.54$ | $9.61 \pm 0.55$ |
| Reweight | $81.14 \pm 0.80$ | $45.84 \pm 0.70$ | $30.37 \pm 1.16$ | $9.75 \pm 0.69$ |
| cIRMv1 | $80.08 \pm 1.90$ | $44.96 \pm 2.88$ | $30.06 \pm 2.07$ | $9.64 \pm 0.94$ |
| cREx | $81.50 \pm 0.76$ | $45.44 \pm 0.96$ | $29.12 \pm 0.97$ | $9.17 \pm 0.59$ |
| cGroupDRO | $78.25 \pm 0.31$ | $41.69 \pm 0.08$ | $28.16 \pm 0.91$ | $9.45 \pm 0.22$ |
| cMMD | $79.64 \pm 0.73$ | $\mathbf{49.44 \pm 0.99}$ | $35.86 \pm 0.66$ | $9.80 \pm 0.45$ |
| PGI | $75.00 \pm 0.85$ | $46.10 \pm 0.79$ | $\mathbf{36.25 \pm 0.42}$ | $\mathbf{11.12 \pm 0.85}$ |
| cMMD (oracle split) | $\mathbf{75.05 \pm 0.98}$ | $47.88 \pm 1.03$ | $37.40 \pm 1.07$ | $10.76 \pm 0.61$ |
| PGI (oracle split) | $70.63 \pm 0.48$ | $\mathbf{48.11 \pm 0.82}$ | $\mathbf{42.69 \pm 0.84}$ | $\mathbf{12.56 \pm 1.20}$ |

Table 5: Hyper-parameters with different validation sets for COLOURED MNIST

| Validation | In-distribution | Non-systematic shift | Systematic shift | Anomaly detection |
|------------|-----------------|---------------------|------------------|-------------------|
| NS+S (PGI) | $99.05 \pm 0.08$ | $98.58 \pm 0.06$ | $98.48 \pm 0.05$ | $89.42 \pm 1.95$ |
| NS (PGI) | $99.31 \pm 0.05$ | $98.21 \pm 0.26$ | $97.54 \pm 0.41$ | $76.00 \pm 4.06$ |
| NS+ID (PGI) | $99.30 \pm 0.07$ | $98.31 \pm 0.27$ | $97.48 \pm 0.45$ | $76.07 \pm 5.67$ |
| ID only (PGI) | $99.69 \pm 0.03$ | $63.62 \pm 2.05$ | $58.18 \pm 2.05$ | $11.81 \pm 1.89$ |
| Base (ERM) | $99.60 \pm 0.02$ | $53.26 \pm 1.89$ | $38.72 \pm 2.27$ | $7.70 \pm 0.23$ |

NS: Hyper-parameters are picked using only the validation set for non-systematic distributional shift (which consists of backgrounds that are different from those in the training set and test sets). This models the situation where we have access to some data that is different from our training data, and is also considered somewhat representative of any shifts we might encounter.

NS + ID: Hyper-parameters are picked using an (equally-weighted) average of the NS and the in-distribution validation sets. If we have prior knowledge of the likelihood of encountering data from out-distributions in the wild, we could use this prior to use an appropriately sampled validation set for hyper-parameter optimisation.

ID ONLY: Hyper-parameters are picked using only the in-distribution validation set.

We show results for the different schemes for our method in Tables 5, 6, 7. While the accuracies under distributional shift are, as expected, less strong than in the previous set of results (NS+S in the tables), we still find improvements over the reference model, indicating that one can still achieve an improved classifier.

In Appendix D, we show similar results with all methods, and include only the best performing method for both generalisation under systematic and non-systematic shift corresponding to the different validation strategies in the tables in this section.

Table 6: Hyper-parameters with different validation sets for COCO-ON-COLOURS

| Validation | In-distribution | Non-systematic shift | Systematic shift | Anomaly detection |
|------------|-----------------|---------------------|------------------|-------------------|
| NS+S (PGI) | $78.23 \pm 2.01$ | $55.57 \pm 4.60$ | $51.62 \pm 3.09$ | $18.84 \pm 2.11$ |
| NS (PGI) | $85.78 \pm 1.45$ | $51.02 \pm 2.32$ | $38.85 \pm 2.29$ | $15.71 \pm 3.25$ |
| NS+ID (PGI) | $85.78 \pm 1.45$ | $51.02 \pm 2.32$ | $38.85 \pm 2.29$ | $15.71 \pm 3.25$ |
| ID only (cMMD) | $92.51 \pm 0.41$ | $44.59 \pm 3.28$ | $10.48 \pm 0.98$ | $6.05 \pm 0.23$ |
| Base (ERM) | $90.57 \pm 1.28$ | $26.81 \pm 4.93$ | $1.10 \pm 0.36$ | $5.47 \pm 0.08$ |

Table 7: Hyper-parameters with different validation sets for COCO-ON-PLACES

| Validation | In-distribution | Non-systematic shift | Systematic shift | Anomaly detection |
|---|---|---|---|---|
| NS+S (cMMD) | $79.64 \pm 0.73$ | $49.44 \pm 0.99$ | $35.86 \pm 0.66$ | $9.80 \pm 0.45$ |
| NS (cMMD) | $79.64 \pm 0.73$ | $49.44 \pm 0.99$ | $35.86 \pm 0.66$ | $9.80 \pm 0.45$ |
| NS+ID (cMMD) | $79.64 \pm 0.73$ | $49.44 \pm 0.99$ | $35.86 \pm 0.66$ | $9.80 \pm 0.45$ |
| ID only (PGI) | $80.99 \pm 0.52$ | $47.63 \pm 0.90$ | $31.91 \pm 0.89$ | $9.59 \pm 0.89$ |
| Base (ERM) | $81.06 \pm 1.01$ | $45.25 \pm 0.96$ | $29.18 \pm 1.24$ | $9.21 \pm 0.21$ |

## 6 CONCLUSION

Our experiments investigate the potential usefulness of invariance penalties and methods at improving performance under distributional shift, such as systematic generalisation and semantic anomaly detection.

While our exploratory experiments are conducted in disambiguated synthetic setups, next steps would involve investigating the potential for extending these approaches to real datasets used in the field. Since such methods cannot work when spurious correlations are completely pervasive, it is important to include sufficient diversity of data sources and curation in order to be able to reap the advantages such techniques can afford us in real world applications. We note that peculiarities in datasets and problems might give rise to different potential failings at robustness, calling for more targeted invariance methods.

We find that our method of learning features that result in matched predictive behaviour throughout training appears to hold promise at handling certain distributional shifts, although it does not always perform best across different validation schemes. A practical line of inquiry would be the question of how to make trade-offs in performance between in-distribution and unexpected situations.

## ACKNOWLEDGMENTS

We thank Ishaan Gulrajani, Tim Cooijmans, David Krueger, and Phong Nguyen for useful discussions, and anonymous reviewers for constructive feedback. We acknowledge the computational resources provided by Compute Canada and Mila, and the financial support of Hitachi, Microsoft Research, CIFAR, and the Natural Sciences and Engineering Research Council of Canada (NSERC Discovery Grant).

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

# A   DATASET DETAILS

In this section, we provide more details about how we constructed our synthetic datasets.

## A.1   COLOURED MNIST

The training set, $T_r$, is constructed with an 80% colour-digit correlation per digit with nine RGB-colours (with the zero digit held out, for testing semantic anomaly detection).

| | |
|---|---|
| 1 | (0,100,0) |
| 2 | (188, 143, 143) |
| 3 | (255, 0, 0) |
| 4 | (255, 215, 0) |
| 5 | (0, 255, 0) |
| 6 | (65, 105, 225) |
| 7 | (0, 225, 225) |
| 8 | (0, 0, 255) |
| 9 | (255, 20, 147) |

Table 8: RGB codes used to bias the digits in the majority group.

The ten colours for the minority group were picked such that their L2 distance is at least 50 units away from the biasing colours. Prior to colouring, the digits were binarised to avoid grayscale tones potentially resulting in unintentionally similar colours.

For the non-systematic validation and test sets, ten colours each were chosen such that they were at least 50 units away from all other colours.

## A.2   COCO-ON-COLOURS

We use the following nine categories for in-distribution objects: *boat*, *airplane*, *truck*, *dog*, *zebra*, *horse*, *bird*, *train*, and *bus*. We hold out *motorcycle* for anomaly detection experiments. For background colours, we use the same colours from the coloured MNIST experiments, and also use an 80/20 split for the majority and minority groups.

In case of multiple instances of the same object in an image, we pick the largest one, and filter our dataset by mask area, such that only images with objects occupying at least 10K pixels are retained. All images are finally resized to $64 \times 64$.

The training set uses 800 such pictures per category, and the validation and test sets use 100 each. The colour backgrounds for the minority group, non-systematically shifted validation and test sets are picked using the same strategy as with the COLOURED MNIST dataset.

## A.3   COCO-ON-PLACES

This dataset follows the same procedure as COCO-ON-COLOURS, except using scenes from the Places dataset. In Table 9 we list the backgrounds from the corresponding scenes for the different categories.

# B   NETWORK ARCHITECTURES AND TRAINING DETAILS

## B.1   COLOURED MNIST

We use a 4-layer CNN with the first three layers being convolutional and the last layer linear. The convolutional layers have feature dimensions of $64 - 128 - 256$, and are all followed by a MAX POOL, BATCH NORM layer, and RELU activation. Before being fed into the final linear layer, there is a spatial mean-pooling operation. An L2 weight decay is added to all parameters with a co-efficient of $1e-4$.

| | Majority group | Minority group | Validation | Test |
|---|---|---|---|---|
| boat | beach | kasbah | oast house | water tower |
| airplane | canyon | lighthouse | orchard | waterfall |
| truck | building facade | pagoda | viaduct | zen garden |
| dog | staircase | rock arch | | |
| zebra | desert (sand) | | | |
| horse | crevasse | | | |
| bird | bamboo forest | | | |
| train | broadleaf forest | | | |
| bus | ball pit | | | |

Table 9: Background scenes for the in-distribution majority group, minority group, and the non-systematically shifted validation and test sets. (The mapping to categories only applies to the majority group in the training set.)

Training is conducted for 30 epochs, with SGD + Momentum (0.9), using batch sizes of 512. The learning rate is cut by 10 from its initial value of 0.1 at epochs 9, 18, and 24.

## B.2   COCO-ON-BACKGROUNDS

For both COCO datasets, we use an architecture based off of Wide Resnet 28-10 (Zagoruyko & Komodakis, 2016). Since our images are $64 \times 64$, we append an extra group of 4 residual blocks with the same layer widths as in the previous group, and use a smaller widening factor of 4 instead of 10 to avoid memory overflow (starting base dimension $= 64$). An L2 weight decay regulariser is applied on all parameters with a coefficient of $5e-4$.

We train for 200 epochs with SGD + Momentum (0.9), using batch sizes of 384, with an initial learning rate of 0.1 which is cut by 10 at the 120th, 160th, 180th, and 190th epochs. We use the initially large learning rate for longer following prior works such as Li et al. (2019) that have suggested annealing schedules with longer periods of higher learning rates can improve generalisation, which we do find to help the base network. In both cases, we apply data augmentation of random crops (after symmetric padding) and random horizontal reflections.

## B.3   PARTITIONING NETWORK

We use the same MLP with three hidden layers for all our partitioning networks, with dimensions $64 - 32 - 16$. We use LAYER NORM (Ba et al., 2016) and RELU activations after each layer. To avoid merely memorising hard examples, it is necessary to regularise this network, so we also apply spectral normalisation (Yoshida & Miyato, 2017); this involves spectrally normalising every linear layer, and excluding the scaling term in the layer normalisation transforms, as in Miyato et al. (2018).

We use a separate network for each class, training for 100 iterations each, with the same batch size as used for training the rest of the model. We use the Adam optimiser (Kingma & Ba, 2014) with a learning rate of $1e-4$. In preliminary experiments we found a shared network for all categories to also work, using conditional layer normalisation (Ba et al., 2016; de Vries et al., 2017). We didn't investigate it further for all datasets, since in general a larger number of classes in a dataset might require larger capacity in the partition predictor to account for more features, and as the number of classes go up, a number of smaller matrices can have a lower footprint than one very large matrix.

Network architecture design for the partitioning network was done only on the COLOURED MNIST dataset, with access to true oracle group labels for a smaller set of in-distribution validation images (20 per category). The same network architecture was applied for the two COCO datasets. The $\gamma$ hyper-parameter was learned separately for all datasets, using the smaller sets of validation images. We find, in preliminary experiments, that using random partitionings lead to much worse performance. This suggests that, although not performed for our present study, in more realistic situations one could potentially tune these hyper-parameters by validating over classification accuracy as for the invariance penalties.

### B.4 Invariance penalties

In all cases, we pause training of the base network after 1 epoch of training, and learn a partitioning of the training set. This learned partition is used to drop in the invariance penalties as training proceeds, and as in prior work (Arjovsky et al., 2019; Krueger et al., 2020), we find ramping in the penalty co-efficient over a number of epochs to be useful for stable training. For IRM and REx (and conditional variants), we find it helpful to scale the ERM term down by the penalty co-efficient when the optimal validation co-efficient is greater than 1, as implemented by Arjovsky et al. (2019) and Krueger et al. (2020).

## C Review of baselines and conditional variants

We briefly review the group invariance methods we compared.

### C.1 IRMv1

In Arjovsky et al. (2019), a risk regularisation method is described in order to encourage reliance on features that obey stable correlations with the target variable across data from different environments. The regularisation consists of a gradient penalty $wrt$ a dummy multiplier on the logits, with the intuition that scaling up or shrinking the logits in different environments can only result in local improvements within each environment if the classifier uses features that correlate at different levels in the different environments. The objective function is

$$\min_{\Phi:\mathcal{X}\to\mathcal{Y}} \sum_{e\in\mathcal{E}} \mathcal{R}^e(\Phi) + \lambda||\nabla_{\mu|\mu=1}\mathcal{R}^e(s.\Phi)||^2. \tag{9}$$

$\Phi$ comprises the predictor, which in our case is $w^\top f_\theta(x)$. $\mu$ is a dummy multiplier, fixed at 1, and $\mathbb{R}^e$ is the environment risk, corresponding to the average loss for data in a particular environment when using $\Phi$.

For our conditional variant (cIRMv1), we stratify the gradient penalty over classes, so that the penalty is applied separately per class in each environment.

The hyper-parameters we search over for this method include the penalty co-efficient $\lambda$ and the number of epochs of training over which to linearly ramp up $\lambda$ to its full value.

### C.2 REx

Krueger et al. (2020) proposed a risk regularisation method that aims to directly match training risks across environments, by imposing a penalty that minimises the variance of risks across environments (V-REx).

$$\min_{\Phi:\mathcal{X}\to\mathcal{Y}} \sum_{e\in\mathcal{E}} \mathcal{R}^e(\Phi) + \lambda\text{Var}(\{\cdots, \mathcal{R}^e, \cdots\}). \tag{10}$$

For our conditional variant (cREx), we apply the variance penalty stratified by class.

The hyper-parameters we search over for this method include the penalty co-efficient $\lambda$ and the number of epochs of training over which to linearly ramp up $\lambda$ to its full value.

### C.3 GroupDRO

Sagawa et al. (2020) suggest an online algorithm for group-based distributionally robust optimisation, which effectively re-weights group losses as a function of their evolving magnitudes, therefore putting more emphasis on groups that fare worse through training.

For our conditional variant (cGroupDRO), we compute the group weights per class, by using the losses belonging to the classes separately in each group.

The hyper-parameters we search over for this method include the learning rate for the online group-weights and the two group adjustment hyper-parameters. Additionally, we sample equally from both groups for this method, as suggested, finding it to improve results in preliminary experiments.

### C.4    REWEIGHT

We learn a hyper-parameter $\lambda$ on validation, such that every example in the majority group is weighted with $1/(\lambda + 1)$ (because we only want to weight the majority group down).

The hyper-parameters we search over for this method include the penalty co-efficient $\lambda$ and the number of epochs of training over which to linearly ramp up $\lambda$ to its full value.

### C.5    MMD FEATURE MATCHING

*Maximum mean discrepancy* based distributional matching of features across domains has been shown to be effective for domain generalisation (Li et al., 2018), and conditional matching of distributions (usually with adversaries, for example, in Li et al. (2018a)) tends to work better. We found in preliminary experiments that conditional MMD significantly outperformed the unconditional variant, so we only ran full experiments and reported results using cMMD.

The group invariance penalty looks as follows

$$||\mathbb{E}\big[\phi(f_\theta(x_{\text{group 0}}))\big] - \mathbb{E}\big[\phi(f_\theta(x_{\text{group 1}}))\big]||^2, \tag{11}$$

where $\phi$ induces a kernel function $K$, which in our implementation is a mixture of 3 Gaussians with bandwidths $[1, 5, 10]$, which are the recommended set of bandwidths in Li et al. (2018). Adding sharper or flatter bandwidths appeared to hurt performance in preliminary experiments.

The hyper-parameters we search over for this method include the penalty co-efficient $\lambda$ and the number of epochs of training over which to linearly ramp up $\lambda$ to its full value.

### C.6    HYPER-PARAMETER GRID SEARCH RANGES

In all cases, $\lambda$ is searched over a range of

$$\{1e{-}4, 1e{-}3, 1e{-}2, 0.1, 0.5, 1, 2, 5, 10, 20, 50, 100, 200, 500, 1000, 10000, 100000\},$$

and the number of epochs over which to linearly ramp up $\lambda$ is searched over $\{1, 5, 30\}$ for MNIST and $\{1, 10, 200\}$ for COCO. For the GroupDRO methods, we search over $\{0.001.0.01, 0.1, 1.0, 10\}$ for the learning rate of the group-weights, and over $\{0, 1, 2, 3, 4, 5\}$ for the group-adjustment hyper-parameters, as recommended in Sagawa et al. (2020). We also average the losses group-wise as already done in IRMv1 and REx for cMMD and PGI, except for COCO-ON-PLACES, where we find this choice to hurt performance.

## D    DIFFERENT VALIDATION SETS

In this section, we report results for all the methods we compare, when picking hyper-parameters using different validation sets, as discussed in Section 5.4.

We note that contrary to what one would typically do in a real-world deployment, we do not augment the training sets with the validation sets for evaluating test time performance. This is because the presence of data with systematic distributional shift at training time improves performance significantly (as observed in Table 1), and our goal here is to perform an illustrative study about the potential effectiveness of invariance methods at learning to generalise systematically.

While we could have augmented the training set with validation data when we are not using validation sets with systematic distributional shift, we follow the same protocol in these cases of not augmenting the training set, in order to keep the numbers comparable with each other across different validation schemes.

Table 10: Picking hyper-parameters only using a validation set of non-systematic shifts for COLOURED MNIST.

| Methods | In-distribution | Non-systematic shift | Systematic shift | Anomaly detection |
|---|---|---|---|---|
| Base (ERM) | $99.60 \pm 0.02$ | $53.26 \pm 1.89$ | $38.72 \pm 2.27$ | $7.70 \pm 0.23$ |
| IRMv1 | $\mathbf{99.61 \pm 0.05}$ | $63.80 \pm 3.58$ | $55.38 \pm 1.52$ | $10.35 \pm 0.43$ |
| REx | $98.95 \pm 0.11$ | $72.12 \pm 1.90$ | $71.18 \pm 3.27$ | $15.54 \pm 2.05$ |
| GroupDRO | $98.70 \pm 0.10$ | $71.51 \pm 2.61$ | $77.95 \pm 0.65$ | $18.26 \pm 2.11$ |
| Reweight | $99.06 \pm 0.06$ | $77.03 \pm 1.33$ | $83.37 \pm 0.61$ | $17.10 \pm 1.11$ |
| cIRMv1 | $99.36 \pm 0.25$ | $65.78 \pm 3.53$ | $61.09 \pm 5.30$ | $14.16 \pm 2.12$ |
| cREx | $99.20 \pm 0.10$ | $73.97 \pm 1.07$ | $76.06 \pm 1.71$ | $17.62 \pm 2.29$ |
| cGroupDRO | $97.89 \pm 0.29$ | $73.71 \pm 3.21$ | $76.90 \pm 2.55$ | $20.73 \pm 4.63$ |
| cMMD | $99.40 \pm 0.07$ | $97.36 \pm 0.72$ | $\mathbf{97.91 \pm 0.19}$ | $\mathbf{78.14 \pm 3.79}$ |
| PGI | $99.31 \pm 0.05$ | $\mathbf{98.21 \pm 0.26}$ | $97.54 \pm 0.41$ | $76.00 \pm 4.06$ |

Table 11: Picking hyper-parameters using both a validation set of non-systematic shifts and the in-distribution set for COLOURED MNIST.

| Methods | In-distribution | Non-systematic shift | Systematic shift | Anomaly detection |
|---|---|---|---|---|
| Base (ERM) | $99.60 \pm 0.02$ | $53.26 \pm 1.89$ | $38.72 \pm 2.27$ | $7.70 \pm 0.23$ |
| IRMv1 | $\mathbf{99.61 \pm 0.05}$ | $63.80 \pm 3.58$ | $55.38 \pm 1.52$ | $10.35 \pm 0.43$ |
| REx | $98.95 \pm 0.11$ | $72.12 \pm 1.90$ | $71.18 \pm 3.27$ | $15.54 \pm 2.05$ |
| GroupDRO | $98.70 \pm 0.10$ | $71.51 \pm 2.61$ | $77.95 \pm 0.65$ | $18.26 \pm 2.11$ |
| Reweight | $99.06 \pm 0.06$ | $77.03 \pm 1.33$ | $83.37 \pm 0.61$ | $17.10 \pm 1.11$ |
| cIRMv1 | $99.36 \pm 0.25$ | $65.78 \pm 3.53$ | $61.09 \pm 5.30$ | $14.16 \pm 2.12$ |
| cREx | $99.20 \pm 0.10$ | $73.97 \pm 1.07$ | $76.06 \pm 1.71$ | $17.62 \pm 2.29$ |
| cGroupDRO | $97.89 \pm 0.29$ | $73.71 \pm 3.21$ | $76.90 \pm 2.55$ | $20.73 \pm 4.63$ |
| cMMD | $99.49 \pm 0.04$ | $96.36 \pm 0.53$ | $\mathbf{97.68 \pm 0.17}$ | $71.15 \pm 2.65$ |
| PGI | $99.30 \pm 0.07$ | $\mathbf{98.31 \pm 0.27}$ | $97.48 \pm 0.45$ | $\mathbf{76.07 \pm 5.67}$ |

Table 12: Picking hyper-parameters using only the in-distribution set for COLOURED MNIST.

| Methods | In-distribution | Non-systematic shift | Systematic shift | Anomaly detection |
|---|---|---|---|---|
| Base (ERM) | $99.60 \pm 0.02$ | $53.26 \pm 1.89$ | $38.72 \pm 2.27$ | $7.70 \pm 0.23$ |
| IRMv1 | $99.69 \pm 0.02$ | $60.18 \pm 1.34$ | $53.20 \pm 1.44$ | $9.71 \pm 0.76$ |
| REx | $\mathbf{99.71 \pm 0.04}$ | $60.71 \pm 1.38$ | $50.87 \pm 2.79$ | $10.02 \pm 0.69$ |
| GroupDRO | $99.61 \pm 0.01$ | $52.21 \pm 2.03$ | $40.27 \pm 2.08$ | $7.37 \pm 0.44$ |
| Reweight | $99.66 \pm 0.04$ | $63.36 \pm 4.60$ | $58.09 \pm 0.52$ | $11.41 \pm 0.49$ |
| cIRMv1 | $99.69 \pm 0.01$ | $60.43 \pm 2.71$ | $52.98 \pm 2.14$ | $10.40 \pm 0.91$ |
| cREx | $99.70 \pm 0.02$ | $61.06 \pm 1.20$ | $50.83 \pm 2.33$ | $9.21 \pm 0.97$ |
| cGroupDRO | $99.63 \pm 0.01$ | $55.53 \pm 3.63$ | $45.25 \pm 2.24$ | $8.69 \pm 1.02$ |
| cMMD | $99.70 \pm 0.02$ | $61.10 \pm 1.66$ | $51.06 \pm 1.87$ | $9.62 \pm 1.09$ |
| PGI | $99.69 \pm 0.03$ | $\mathbf{63.62 \pm 2.05}$ | $\mathbf{58.18 \pm 2.05}$ | $\mathbf{11.81 \pm 1.89}$ |

Table 13: Picking hyper-parameters only using a validation set of non-systematic shifts for COCO-ON-COLOURS.

| Methods | In-distribution | Non-systematic shift | Systematic shift | Anomaly detection |
|---|---|---|---|---|
| Base (ERM) | $90.57 \pm 1.28$ | $26.81 \pm 4.93$ | $1.10 \pm 0.36$ | $5.47 \pm 0.08$ |
| IRMv1 | $91.61 \pm 0.38$ | $32.30 \pm 4.52$ | $2.11 \pm 0.30$ | $5.81 \pm 0.17$ |
| REx | $\mathbf{91.69 \pm 0.50}$ | $36.57 \pm 4.03$ | $2.69 \pm 0.81$ | $5.73 \pm 0.14$ |
| GroupDRO | $40.31 \pm 2.11$ | $38.84 \pm 3.78$ | $\mathbf{43.24 \pm 2.84}$ | $17.99 \pm 3.68$ |
| Reweight | $73.17 \pm 2.48$ | $48.98 \pm 2.65$ | $39.80 \pm 2.61$ | $18.20 \pm 3.80$ |
| cIRMv1 | $91.53 \pm 0.31$ | $31.11 \pm 4.51$ | $1.74 \pm 0.40$ | $5.87 \pm 0.16$ |
| cREx | $91.45 \pm 0.39$ | $32.43 \pm 2.03$ | $1.98 \pm 0.68$ | $5.75 \pm 0.13$ |
| cGroupDRO | $43.61 \pm 4.33$ | $39.15 \pm 4.79$ | $36.63 \pm 4.81$ | $\mathbf{18.21 \pm 3.65}$ |
| cMMD | $89.87 \pm 1.13$ | $\mathbf{55.02 \pm 2.29}$ | $27.36 \pm 1.57$ | $8.82 \pm 0.70$ |
| PGI | $85.78 \pm 1.45$ | $51.02 \pm 2.32$ | $38.85 \pm 2.29$ | $15.71 \pm 3.25$ |

Table 14: Picking hyper-parameters using both a validation set of non-systematic shifts and the in-distribution set for COCO-ON-COLOURS.

| Methods | In-distribution | Non-systematic shift | Systematic shift | Anomaly detection |
|---|---|---|---|---|
| Base | $90.57 \pm 1.28$ | $26.81 \pm 4.93$ | $1.10 \pm 0.36$ | $5.47 \pm 0.08$ |
| IRMv1 | $91.61 \pm 0.38$ | $32.30 \pm 4.52$ | $2.11 \pm 0.30$ | $5.81 \pm 0.17$ |
| REx | $\mathbf{91.69 \pm 0.50}$ | $36.57 \pm 4.03$ | $2.69 \pm 0.81$ | $5.73 \pm 0.14$ |
| GroupDRO | $90.70 \pm 0.56$ | $33.10 \pm 3.26$ | $5.66 \pm 0.95$ | $6.60 \pm 0.40$ |
| Reweight | $90.25 \pm 0.71$ | $40.23 \pm 3.32$ | $10.60 \pm 1.34$ | $7.06 \pm 0.52$ |
| cIRMv1 | $91.53 \pm 0.31$ | $31.11 \pm 4.51$ | $1.74 \pm 0.40$ | $5.87 \pm 0.16$ |
| cREx | $91.45 \pm 0.39$ | $32.43 \pm 2.03$ | $1.98 \pm 0.68$ | $5.75 \pm 0.13$ |
| cGroupDRO | $87.68 \pm 0.59$ | $36.40 \pm 2.30$ | $14.07 \pm 2.47$ | $9.82 \pm 0.91$ |
| cMMD | $89.87 \pm 1.13$ | $\mathbf{55.02 \pm 2.29}$ | $27.36 \pm 1.57$ | $8.82 \pm 0.70$ |
| PGI | $85.78 \pm 1.45$ | $51.02 \pm 2.32$ | $\mathbf{38.85 \pm 2.29}$ | $\mathbf{15.71 \pm 3.25}$ |

Table 15: Picking hyper-parameters using only the in-distribution set for COCO-ON-COLOURS.

| Methods | In-distribution | Non-systematic shift | Systematic shift | Anomaly detection |
|---|---|---|---|---|
| Base | $90.57 \pm 1.28$ | $26.81 \pm 4.93$ | $1.10 \pm 0.36$ | $5.47 \pm 0.08$ |
| IRMv1 | $91.54 \pm 0.37$ | $32.40 \pm 3.62$ | $1.93 \pm 0.36$ | $5.77 \pm 0.23$ |
| REx | $91.62 \pm 0.38$ | $31.89 \pm 4.08$ | $1.98 \pm 0.37$ | $5.74 \pm 0.20$ |
| GroupDRO | $91.44 \pm 0.27$ | $22.42 \pm 3.00$ | $0.56 \pm 0.15$ | $5.55 \pm 0.19$ |
| Reweight | $91.10 \pm 0.50$ | $38.63 \pm 3.23$ | $4.35 \pm 1.13$ | $\mathbf{6.13 \pm 0.22}$ |
| cIRMv1 | $91.31 \pm 0.43$ | $30.94 \pm 3.73$ | $1.65 \pm 0.36$ | $5.83 \pm 0.17$ |
| cREx | $91.70 \pm 0.50$ | $34.93 \pm 4.58$ | $2.24 \pm 0.48$ | $5.82 \pm 0.19$ |
| cGroupDRO | $91.75 \pm 0.60$ | $24.05 \pm 3.44$ | $0.94 \pm 0.27$ | $5.77 \pm 0.13$ |
| cMMD | $92.51 \pm 0.41$ | $\mathbf{44.59 \pm 3.28}$ | $\mathbf{10.48 \pm 0.98}$ | $6.05 \pm 0.23$ |
| PGI | $\mathbf{91.86 \pm 0.33}$ | $32.46 \pm 3.06$ | $2.81 \pm 0.53$ | $5.88 \pm 0.19$ |

Table 16: Picking hyper-parameters only using a validation set of non-systematic shifts for COCO-ON-PLACES.

| Methods | In-distribution | Non-systematic shift | Systematic shift | Anomaly detection |
|---|---|---|---|---|
| Base (ERM) | $81.06 \pm 1.01$ | $45.25 \pm 0.96$ | $29.18 \pm 1.24$ | $9.21 \pm 0.21$ |
| IRMv1 | $80.93 \pm 0.71$ | $45.17 \pm 0.92$ | $28.78 \pm 0.73$ | $9.39 \pm 0.60$ |
| REx | $81.25 \pm 0.76$ | $45.40 \pm 0.95$ | $29.20 \pm 1.28$ | $9.46 \pm 0.98$ |
| GroupDRO | $76.05 \pm 0.87$ | $43.72 \pm 0.43$ | $31.83 \pm 0.54$ | $9.61 \pm 0.55$ |
| Reweight | $80.90 \pm 0.50$ | $44.87 \pm 1.26$ | $29.34 \pm 0.99$ | $9.59 \pm 0.54$ |
| cIRMv1 | $81.48 \pm 0.67$ | $45.59 \pm 1.27$ | $29.28 \pm 0.96$ | $9.80 \pm 0.78$ |
| cREx | $\mathbf{81.50 \pm 0.76}$ | $45.44 \pm 0.96$ | $29.12 \pm 0.97$ | $9.17 \pm 0.59$ |
| cGroupDRO | $78.25 \pm 0.31$ | $41.69 \pm 0.08$ | $28.16 \pm 0.91$ | $9.45 \pm 0.22$ |
| cMMD | $79.64 \pm 0.73$ | $\mathbf{49.44 \pm 0.99}$ | $\mathbf{35.86 \pm 0.66}$ | $\mathbf{9.80 \pm 0.45}$ |
| PGI | $80.99 \pm 0.52$ | $47.63 \pm 0.90$ | $31.91 \pm 0.89$ | $9.59 \pm 0.89$ |
| cMMD (oracle split) | $\mathbf{80.04 \pm 1.01}$ | $\mathbf{49.02 \pm 1.18}$ | $35.60 \pm 0.72$ | $10.55 \pm 0.55$ |
| PGI (oracle split) | $75.98 \pm 0.75$ | $47.50 \pm 0.87$ | $\mathbf{37.27 \pm 1.40}$ | $\mathbf{11.57 \pm 0.71}$ |

Table 17: Picking hyper-parameters using both a validation set of non-systematic shifts and the in-distribution set for COCO-ON-PLACES.

| Methods | In-distribution | Non-systematic shift | Systematic shift | Anomaly detection |
|---|---|---|---|---|
| Base (ERM) | $81.06 \pm 1.01$ | $45.25 \pm 0.96$ | $29.18 \pm 1.24$ | $9.21 \pm 0.21$ |
| IRMv1 | $80.93 \pm 0.71$ | $45.17 \pm 0.92$ | $28.78 \pm 0.73$ | $9.39 \pm 0.60$ |
| REx | $81.25 \pm 0.76$ | $45.40 \pm 0.95$ | $29.20 \pm 1.28$ | $9.46 \pm 0.98$ |
| GroupDRO | $80.61 \pm 0.44$ | $41.96 \pm 1.00$ | $27.19 \pm 0.67$ | $9.05 \pm 0.06$ |
| Reweight | $80.90 \pm 0.50$ | $44.87 \pm 1.26$ | $29.34 \pm 0.99$ | $9.59 \pm 0.54$ |
| cIRMv1 | $81.48 \pm 0.67$ | $45.59 \pm 1.27$ | $29.28 \pm 0.96$ | $9.80 \pm 0.78$ |
| cREx | $\mathbf{81.50 \pm 0.76}$ | $45.44 \pm 0.96$ | $29.12 \pm 0.97$ | $9.17 \pm 0.59$ |
| cGroupDRO | $78.25 \pm 0.31$ | $41.69 \pm 0.08$ | $28.16 \pm 0.91$ | $9.45 \pm 0.22$ |
| cMMD | $79.64 \pm 0.73$ | $\mathbf{49.44 \pm 0.99}$ | $\mathbf{35.86 \pm 0.66}$ | $\mathbf{9.80 \pm 0.45}$ |
| PGI | $80.99 \pm 0.52$ | $47.63 \pm 0.90$ | $31.91 \pm 0.89$ | $9.59 \pm 0.89$ |
| cMMD (oracle split) | $\mathbf{79.56 \pm 0.64}$ | $46.74 \pm 0.83$ | $\mathbf{34.78 \pm 0.76}$ | $9.78 \pm 0.59$ |
| PGI (oracle split) | $78.70 \pm 0.86$ | $\mathbf{47.28 \pm 1.05}$ | $32.84 \pm 0.89$ | $\mathbf{11.13 \pm 0.90}$ |

Table 18: Picking hyper-parameters using only the in-distribution set for COCO-ON-PLACES.

| Methods | In-distribution | Non-systematic shift | Systematic shift | Anomaly detection |
|---|---|---|---|---|
| Base (ERM) | $81.06 \pm 1.01$ | $45.25 \pm 0.96$ | $29.18 \pm 1.24$ | $9.21 \pm 0.21$ |
| IRMv1 | $80.93 \pm 0.71$ | $45.17 \pm 0.92$ | $28.78 \pm 0.73$ | $9.39 \pm 0.60$ |
| REx | $81.25 \pm 0.76$ | $45.40 \pm 0.95$ | $29.20 \pm 1.28$ | $9.46 \pm 0.98$ |
| GroupDRO | $80.61 \pm 0.44$ | $41.96 \pm 1.00$ | $27.19 \pm 0.67$ | $9.05 \pm 0.06$ |
| Reweight | $\mathbf{81.53 \pm 0.66}$ | $45.77 \pm 1.33$ | $29.39 \pm 0.97$ | $9.55 \pm 0.79$ |
| cIRMv1 | $81.48 \pm 0.67$ | $45.59 \pm 1.27$ | $29.28 \pm 0.96$ | $9.80 \pm 0.78$ |
| cREx | $80.68 \pm 0.69$ | $44.80 \pm 1.39$ | $29.76 \pm 1.05$ | $\mathbf{9.95 \pm 0.79}$ |
| cGroupDRO | $80.23 \pm 0.13$ | $41.86 \pm 0.60$ | $25.88 \pm 1.20$ | $9.43 \pm 0.68$ |
| cMMD | $81.11 \pm 0.51$ | $46.57 \pm 0.97$ | $31.54 \pm 0.88$ | $9.79 \pm 0.79$ |
| PGI | $80.99 \pm 0.52$ | $\mathbf{47.63 \pm 0.90}$ | $\mathbf{31.91 \pm 0.89}$ | $9.59 \pm 0.89$ |
| cMMD (oracle split) | $\mathbf{81.59 \pm 0.65}$ | $\mathbf{45.47 \pm 1.40}$ | $29.16 \pm 0.96$ | $9.15 \pm 0.36$ |
| PGI (oracle split) | $81.22 \pm 1.09$ | $45.16 \pm 0.96$ | $\mathbf{29.24 \pm 0.64}$ | $\mathbf{9.31 \pm 0.67}$ |

## E    MEASURING SEMANTIC ANOMALY DETECTION

We use the test set with non-systematic distributional shift as the normal data, and the held-out class data combined systematically with the biasing colours or backgrounds as the anomalous data. For MNIST, this means there is 9 times more normal data than anomalous data, which reflects the typical situation of anomalies being rarer. Our choice of normal data makes this a harder task than usual, since we are assessing for higher (than the anomalies) predictive confidences for non-semantic shift with semantic factors kept the same, and reduced predictive confidence for semantic shift with non-semantic factors from the seen data. For the COCO datasets, we only sample 100 images from the held-out class to resemble the MNIST experimental setup.

Anomaly detection is measured using average precision, treating the anomalous class as positive, with the negative of the predictive softmax confidence as the score (Hendrycks & Gimpel, 2017).

## F    ALGORITHM

---

**Algorithm 1:** Algorithm for PGI

---

Initialise all classifier parameters $\theta, w$ and partition-predicting networks, $g_c, \forall c \in [C]$ ;
**for** *one epoch* **do**
    **for** *mini-batches* $\mathcal{D}_b \in \mathcal{D}$ **do**
        $grad_\theta := \nabla_\theta \ell(\theta, w | \mathcal{D}_b)$ ;
        $grad_w := \nabla_w \ell(\theta, w | \mathcal{D}_b)$ ;
        $\theta, w := optimizer(grad_\theta, grad_w)$ ;
    **end**
**end**
**for** *all classes* $c \in [C]$ **do**
    Learn a partition for images in $\mathcal{D}$ with labels $c$, (Eq. 8)
**end**
**for** $T - 1$ *epochs* **do**
    **for** *mini-batches* $\mathcal{D}_b \in \mathcal{D}$ **do**
        $grad_\theta := \nabla_\theta (\ell(\theta, w | \mathcal{D}_b) + \lambda.penalty)$ (Eq. 6,7) ;
        $grad_w := \nabla_w \ell(\theta, w | \mathcal{D}_b)$ ;
        $\theta, w := optimizer(grad_\theta, grad_w)$ ;
    **end**
**end**

---

