# OpenReview forum: "Systematic generalisation with group invariant predictions"
_ICLR.cc/2021/Conference — ICLR 2021 Spotlight_

### Official Review · AnonReviewer3 · 2020-10-28

**Rating:** 8
**Confidence:** 3

**Review:**

Summary:
This paper studies the behaviour of deep neural networks in situations where simple but irrelevant correlations exist between input and output, and dominate more complex but relevant correlations. The authors conduct experiments on synthetic datasets (like coloured MNIST) and show that an invariance penalty helps the network focus on relevant correlations.

Pros:
- The paper studies neural network behaviour with respect to systemic biases that are likely faced by most neural networks in some form or the other. To make the study tenable, the authors make use of meaningful synthetic datasets, and propose an intuitive regularization to overcome the systemic biases.
- The analysis done in the paper is very methodical, and the presentation is very clear.
- The numerical simulations are comprehensive and convincing.

Cons:
- It would be nice to see how this would be applicable to real world datasets. The paper is interesting even without it, and I also appreciate that the authors are honest about it - so I would not hold it against the authors. But it would further strengthen the paper if some basic experiments are done on real world datasets. For instance, will one be able to find a partition on ImageNet?

Comments:
- Section 5.1: Minimization is spelt incorrectly.
- Equation (6-7): I am not entirely sure what is happening with respect to the constraint on \theta. What does capital \theta correspond to? And if \theta itself is the result of an optimisation (argmin), then why is there another optimisation on the same \theta in the loss function?
- In the text that appears before equation (3), it is mentioned that the predicted features f_\theta will be matched for the two partitions, but equation (7) matches the predicted output post softmax. Could you please clarify?

---

> ### Author Response · Authors · 2020-11-17
> **Response to review**
>
> Thanks for the review!
>
> Re. more realistic data, early experiments on STL-10 indicated that partitions may be found, with some signs that improvements at (in-distribution) generalisation are possible with these partitions. For this particular submission, we did not pursue this direction because we wanted setups corresponding to clear varieties of distribution shifts, and it was difficult to manually identify such situations in the standard test sets. In general, in a typical dataset such as Imagenet (large number of pictures scraped from the internet without very careful curation), there might be many factors that do not correspond very well to the sort of biases we explored (while there are works suggesting background correlations play a significant role: https://arxiv.org/abs/2006.09994). However, while Imagenet tends to be the go-to dataset for computer vision object recognition researchers, it is far from being a dataset which accurately models all practical problems that concern us; in many cases (such as medical imaging for example), one often does not have a rich source of diverse data with many different categories. The question of figuring out the specific robustness issues arising due to particular quirks of datasets and problems is interesting, and we hope to see more investigation along these lines. Such findings can then guide us towards developing more targeted invariance methods.
>
> Re. Eq. (6,7) (now Eq. 4,5 because we removed numbering for two equations), this is a constrained bi-level optimisation problem where the “inner” optimisation is the constraint. Capital $\Theta$ refers to the entire unconstrained space of $\theta$. Note that $\in$ is used instead of equality; this means we wish to restrict the search space of $\theta$ to solutions that satisfy the constraint.
>
> Re. the text above Eq. (3) (now Eq. 1), we are matching the features, but through a frozen last layer which yields the softmaxes; the optimisation for the invariance penalty takes place only over the feature-extractor parameters $\theta$ and not the $w$, which means that only these parameters are trained. We describe this in the text: “the (class-conditioned) distributions of features to match in the sense that they lead to the same softmax distributions on average”.

---

### Official Review · AnonReviewer1 · 2020-10-28
**Good work introducing a simple method to improve systematic generalisation**

**Rating:** 8
**Confidence:** 4

**Review:**

This paper shows that group invariance methods across inferred partitions show better generalization in (non-)systematic distributional shifts and anomaly detection settings. It also suggests a new invariance penalty and empirically shows that it works better on three synthetic datasets viz. coloured-MNIST, COCO-on-colours, and COCO-on-places.

The paper is written well and starts off by giving an intuition of why IRM-like methods are important by presenting the results of a simple experiment on coloured-MNIST (table 1). It then goes on to talk about (non-)systematic generalization before introducing the proposed method. The authors use reverse KL divergence between the group distributions as the penalty and use prior work to partition the datasets into groups. They use

The results look promising across datasets, though it is slightly lower in the 'in-distribution' setting. I am happy to see that they also talk extensively about hyperparameter selection especially in the case where they assume no access to validation sets with a distributional shift.

Overall, I like the work and would like to see it presented at the conference.
One minor point: cite work the first time you introduce something, not later on. It can be a little confusing for the readers. I wondered if I missed something. For ex: "We find that a recently proposed method can be effective at discovering...", "IRMv1", etc.

---

> ### Author Response · Authors · 2020-11-17
> **Response to review**
>
> Thanks for the review! About the citations; we’ve taken a pass with this in mind, fixing the instances we could find.

---

### Official Review · AnonReviewer4 · 2020-10-28
**Well written paper, perhaps a bit incremental improvement but interesting and well executed**

**Rating:** 6
**Confidence:** 3

**Review:**

Summary: the paper studies a setting where there are simple correlations with the target variables (that are not however robust) and more complex but robust features. The simple correlation is usually such that in most cases the feature is descriptive of the label, but at times it takes values that are part of a “minority group” that is not descriptive of any one class in particular.
Systematic-shift generalization is tested using the same spurious features that are present in training, in all combinations except the usual pairing of spurious feature and class.
Non-systematic shifts are tested using novel spurious features.
Anomaly detection is tested with unseen robust features.
Neural networks trained with standard ERM notoriously pick up on all features that correlate with the label, and the paper compares several methods (including IRMv1, REx, and GroupDRO) to the proposed Predictive Group Invariance. The latter is shown to work better than the baselines (even when using class conditioning) on systematic shifts and anomaly detection, and often non-systematic shifts.

I found the paper to be mostly very well written (with few exceptions), and especially sections 1 and 2 very easy to read and understand. The experiments setting is clear and not at all trivial.

My main questions and concern:
- The description of PGI could be improved, I would suggest adding an algorithm box that sums up what is described.
- “environments” and “partitions” are the same thing for PGI? How are “environments” chosen for the baseliness that need them? With the same partition networks used for PGI? If not, it might be hard to disentangle the role of using the partition networks from the KL objective of PGI.
- The part about partitioning is not detailed enough. P13: "We use a separate network for each object category” what is an object category? From page 3 I understood that there might be 2 categories (easy and hard), is this the case? Is it correct to think of the role of these partition predicting networks as a sort of clustering that focuses on the most easy-to-find features?
- I haven’t seen any mention of early stopping in the experiments. It sounds like the performance reported is the one at the end of all training epochs. This might explain why for example ERM on COCO-on-Colours only achieves a 1.10% accuracy on Systematic shift. If this is correct, why not using early stopping, given how prone to overfitting these networks might be?

Overall, based on the results, the proposed PGI seems to improve the accuracy over the baselines, even though it seems to be more of an incremental improvement than a substantial and conceptual one.
I look forward to a constructive discussion with the authors and hope my questions and concerns will be clarified.


Minor:
- I think the blanket statement “it has been reported that highly competitive performances can often be achieved with a baseline model on such domain generalisation benchmarks (Gulrajani & Lopez-Paz, 2020), similarly as in Table 1. “ is a bit too vague, and since it questions the validity of the results in all papers mentioned before, it should be either properly justified or adjusted to make sure that what it says actually applies to all of them.
- Figure 3 is referenced in the text instead of Figure 2 (perhaps it was not referenced with \ref )

---

> ### Author Response · Authors · 2020-11-17
> **Response to review**
>
> Thanks for the review! We respond to comments and questions below:
>
>
> > “environments” and “partitions” are the same thing for PGI? How are “environments” chosen for the baselines that need them? With the same partition networks used for PGI?
>
> We use “environments” (which is the term used in some of the recent OoD literature) and “partitions” (in our context of split datasets) synonymously. We see that it can be confusing to use both terms, so we are editing all instances of “environments” to be “partitions” unless we are talking about terminology in the context of existing works, in which case we mention that “environment” refers to a source of data.
> Yes, we use the same partition inferring networks for all invariance methods.
>
> >The part about partitioning is not detailed enough. P13: "We use a separate network for each object category” what is an object category? From page 3 I understood that there might be 2 categories (easy and hard), is this the case? Is it correct to think of the role of these partition predicting networks as a sort of clustering that focuses on the most easy-to-find features?
>
> We refer to classes by “object category”, for example the digits 0, 1, 2, … In section 3, when we index “object categories” by c, we mean the class, and the term “groups” are used to refer to the biased (easy) and unbiased (hard) subsets of data. We’re replacing “object category” with “class” to avoid confusion.
> The partition predictor may be considered to perform a feature-and-loss based clustering in some sense, in that we identify common features that lead to low losses, leading to maximal violations of the invariant learning objective in Arjovsky et al. We also initially tried KMeans based methods and found these to perform poorly compared to this method.
>
> > I haven’t seen any mention of early stopping in the experiments. It sounds like the performance reported is the one at the end of all training epochs. This might explain why for example ERM on COCO-on-Colours only achieves a 1.10% accuracy on Systematic shift.
>
> We did look at early stopping in preliminary experiments, but since our setups are mostly concerned with easily learned non-semantic correlations, we found that systematic generalisation tends to be worse earlier on, since at earlier stages of training the network is still predominantly reliant upon the simpler features it has quickly learned. Early stopping can be useful when networks learn complex non-robust features in later stages of training in order to overfit to the training set. It is possible to achieve around 11% accuracy on COCO-on-Colours if we stop very early, almost immediately after starting training, but at that point the network is not very useful, also achieving ~11% accuracy in-distribution accuracy.
>
> > the blanket statement...questions the validity of the results in all papers mentioned before
>
> We meant to imply that systematic generalisation is not something that has been tested in the way we do before, and that while it has been reported that ERM can perform well at common domain generalisation benchmarks, setups such as ours present a challenge to ERM training. We agree that this might potentially be a contentious comment since it calls into question many previous findings; we’re removing it.
>
> We’ve added an algorithm box in the appendix, and fixed the figure reference, thanks for spotting that!

---

### Official Review · AnonReviewer2 · 2020-10-30
**Reasonable paper, some clarity issues**

**Rating:** 6
**Confidence:** 4

**Review:**

Overall, I found the paper well-written, the methods appear reasonable and the math appears correct. I think the case made for the importance/significance of the method could be improved, but think the paper should be accepted regardless.

--- Comments ---

1. I thought the introduction did a good job setting up the high-level problem, but did not really establish why a new method is needed. I got to the end of the intro and wondered why I couldn't use any one of a dozen DRO type methods to get the type of robustness described. The experiments do a reasonable job of demonstrating the utility of the proposed method, but I recommend adding something to the intro like: Methods have been proposed that promote robustness to distributional shift, however, these methods fail to capture XYZ shifts because ABC.

2. I really liked the examples of the different types of shifts that the authors are interested in, but thought the paper could do a better job arguing why these specific shifts are important or relevant. Perhaps swapping the "red chair" example for something more compelling might do the trick. In particular, it is worth giving a compelling reason why the drop in average performance might be worth an improvement in, for example, anomaly detection.

3. I found the early parts of Section 3 a bit confusing because it was unclear at that point in the paper where the "majority" and "minority" groups were coming from. I recommend adding a few sentences to the beginning of that section like: Suppose that we knew the collection of non-semantic features and thus new the relevant majority and minority groups. This will not, in general, be the case and we will show how to derive such groups from the data, but we first establish our method as if such groups were known.

4. I found Equation (10) very hard to follow. First, the notation has *many* problems (superscripts from earlier in the section are changed to subscripts; no domain is given for $\alpha$; it is not clear what it means to index $\alpha$ by $e$; $\mu$ is not defined; $\gamma$ is not defined). Second, I would give a few sentences explaining what the pieces of this objective are doing and why it achieves the goal of splitting the data into relevant majority and minority groups.

--- Minor comments ---

1. Page 2, par 3, line 1: Not clear what "modeling bias" refers to here or why it affects the dataset.

2. Equations (1) and (2) don't really seem necessary for the rest of the work. In particular, $\mathcal{C}$ isn't reference anywhere else in the paper. I would just define $h_s$ and $h_n$ and move on.

3. I found the $\not\sim$ notation a bit confusing. Does it mean "sampled from any distribution other than $p$? Based on the description, it seems more like it means "sampled from outside the support of $p$" which matches the examples given in Fig 1.

4. Equation 6: $\ell$ is not defined.

5. Equation 9 and surrounding text: I would change reverse-KL to just KL since, in this context, this not an obvious forward/reverse direction.

6. Text under Equation 9: It doesn't really make sense for a categorical distribution to be multi-modal since there is no inherent ordering to values in the support (unless there are literally two or more equivalent modes). I would recommend changing unimodal/multi-modal to peaked/flat or low-entropy/high-entropy.

---

> ### Author Response · Authors · 2020-11-17
> **Response to review**
>
> Thanks for the review! We respond to comments below -
>
> 1. We agree that a more in-depth discussion of why certain methods might be better suited or easier to optimise than others would be much more compelling. We have some intuitions that guided development -- for example, we thought DRO-type methods might hurt in-distribution performance more because they take “less advantage” of all the data due to down-weighting, KL-type losses in the output space would be easier to optimise with the standard cross-entropy -- but these are speculative and we don’t really have a clear theory; our explorations are mainly empirical. We expect that as the conversation continues to evolve around robust models, we shall better identify underlying mechanisms and get closer to figuring out what things can work best for which situations.
> 2. Early in the introduction, we use the red chairs as a quick hook to intuitively motivate learning complex but robust features without details about the different types of robustness. But we agree that it would help to provide clearer motivations for why such predictive rules might be preferred. We are adding the following sentence to the end of Section 2, once the reader has been introduced to the different kinds of generalisations:“Improving performance for such scenarios involving distributional shift might come at a cost for in-distribution performance, since more robust, complex features might be harder to learn than simpler dominant correlations that hold in-distribution. In many real-world applications where one is likely to encounter unexpected situations, such as a self-driving car, it can often be preferable to use such classifiers that can indicate reduced confidence upon encountering anomalous objects, or continue to operate in changing environments.”
> 3. This indeed helps, we’re adding the following sentence to the start of Section 3: “In general, we do not expect to have direct knowledge of majority and minority groups corresponding to the biasing non-semantic features in a dataset. We will later show how one might infer such groups from the data, but we first describe an invariance penalty assuming we have access to the groups.”
> 4. Apologies for the confused notation; we have made the suggested changes - (1) subscripts for indexing examples have been restored to superscripts as before, (2) \alpha has been substituted with \beta described to be a soft-partition, with every element in [0,1], which finally gives the hard-partitions \alpha referred to previously (3) a description has been added for e (which indexes the partition) and \beta^i(e) is described to be the predicted probability for the i-th example being in partition e, such that \beta^i(e=0) + \beta^i(e=1) = 1, (4) \gamma is defined to be a hyper-parameter. Details for why this can work is best explained in the cited papers (Arjovsky et al. and Creager et al.); we are including the following brief sentence to describe the principle at a high level: “The key intuition is that an invariant learning objective, as formulated by IRM \cite{arjovsky2019invariant}, is maximally violated by splitting along a spurious correlation when predictions rely exclusively on it in a reference model (see Theorem 1 in \citet{creager2020eiil} for details)”.
>
> Minor comments:
> 1. We meant to imply that we are modelling bias in datasets with our synthetic setups, but agree this was awkward phrasing. We are editing to simply say “when the dataset is biased”.
> 2. While we don’t directly use (1) and (2) elsewhere, we believe there might be some expository value to writing this down (pointing to possibilities with being able to synthesise data with clearer characterisations of C, for example). But we also agree that this doesn’t need to be made a big deal of in this particular submission; we are moving this notation inline without explicit numbering.
> 3. We’re adding a footnote to say that we mean “sampled from outside the support of p” with the \not \sim notation.
> 4. We’ve added this.
> 5. Agreed, we’ve edited.
> 6. Edited. Thanks a lot for all these suggested improvements!

---

### Comment · ~Xavier_Boix1 · 2021-03-10
**Code**

Congratulations on the acceptance of the paper!

We are having trouble reproducing parts of this paper. Do the authors plan to make the code publicly available?

---

> ### Comment · ~Faruk_Ahmed2 · 2021-03-12
> **Code made available**
>
> Thanks! We've put up code at https://github.com/Faruk-Ahmed/syst-gen

---

> > ### Comment · ~Xavier_Boix1 · 2021-03-15
> > **Thanks!!**
> >
> > Thank you so much! We appreciate your prompt reply.

---

### Decision · Program_Chairs · 2021-01-07
**Final Decision**

**Decision:**

Accept (Spotlight)

**Comment:**

All reviewers seems in favour of accepting this paper, witht he majority voting for marginally above acceptance threshold.
The authors have taken special heed of the suggestions and improved the clarity of the paper.
From examination of the reviews, the paper achieves enough to warrant publication.
My recommendation is therefore to accept the manuscript.